# Decrease of Hyaluronidase Activity and Suppression of Mouse CD4+ T Lymphocyte Activation by Tomato Juice Saponin Esculeoside B, and Its Sapogenol Esculeogenin B

**DOI:** 10.3390/jpm12040579

**Published:** 2022-04-05

**Authors:** Jian-Rong Zhou, Nozomi Kitahara, Honami Nakamura, Takuya Ono, Ryohei Karashima, Jun Fang, Toshihiro Nohara, Kazumi Yokomizo

**Affiliations:** Faculty of Pharmaceutical Sciences, Sojo University, Kumamoto 860-0082, Japan; g1151034@m.sojo-u.ac.jp (N.K.); g1151124@m.sojo-u.ac.jp (H.N.); g1251024@m.sojo-u.ac.jp (T.O.); g1251028@m.sojo-u.ac.jp (R.K.); fangjun@ph.sojo-u.ac.jp (J.F.); none@ph.sojo-u.ac.jp (T.N.); yoko0514@ph.sojo-u.ac.jp (K.Y.)

**Keywords:** CD4+ T lymphocyte, esculeogenin B (Esg-B), esculeoside B (EsB), Th1, Th2, tomato juice/canned saponin, Treg

## Abstract

(1) Background: A naturally occurring glycoside, esculeoside B (EsB), has been identified as a major component in juice or canned tomato. We reported how EsB ameliorated mice experimental atopic dermatitis by a decrease in serum IgE levels. However, the underlying immunologic molecular mechanisms are unknown. (2) Methods: The present study tested the effects of EsB on hyaluronidase activity and CD4+ T lymphocyte activation using concanavalin A (ConA)-blast mouse splenocyte primary culture. (3) Results: We found that EsB and its sapogenol esculeogenin B (Esg-B) decreased hyaluronidase activity by a modified Morgan–Elson method. We demonstrated that EsB/Esg-B dose-dependently suppressed T-lymphoproliferation using CFSE-labeled flow-cytometry and water-soluble tetrazolium (WST) assay. Using ELISA and q-PCR methods, EsB/Esg-B suppressed the cytokine secretion and mRNA expression of Th2-relevant IL-4 and Th1-relevant IFN-γ. Moreover, both EsB/Esg-B showed a reduction in IL-10 secretion, but only Esg-B decreased IL-2 secretion. (4) Conclusions: Our study is the first to demonstrate how EsB/Esg-B inhibit hyaluronidase activity and reduce CD4+ T-lymphocyte activation via a reduction in Th2-lymphocyte activity by modulation of Th2/Th1/Treg subunits differentiation.

## 1. Introduction

Atopic dermatitis (AD) is a chronic inflammatory disorder predominantly mediated by CD4+ T helper cells [1,2,3,4]. First, an overexpression of Th2 cell cytokines was observed in acute and chronic lesions of AD. The Th2 response is characterized by skewed cytokine production including IL-4 and IL-5, and associated downstream GATA binding protein 3 (GATA3), which in turn drive eosinophilic inflammation and immunoglobulin class switching to IgE in B cells and subsequent release of inflammatory mediators [5]. Second, it has been reported that chronic lesions also express moderate levels of the Th1 cell cytokine IFN-γ; thus, the T cell response present in AD is not only Th2-polarized but may lead to heterogeneous cytokine production involving Th1 cell cytokines [6]. Third, it has been reported that children with AD had higher Th2, memory Regulatory T (Treg) cell numbers compared to healthy children. While higher memory Treg cells represent chronic inflammation, naive cells have no memory of sensitization to specific antigens [7,8,9]. The major population of CD4+ Treg cells was found to be characterized by high expression of IL-2 receptor alpha chain (CD25) and transcription factor Foxp3 (forkhead box P3), which are important for the development and suppressive function of Treg cells.

Moreover, it is well know that hyaluronidase, an enzyme for hyaluronic acid (HA) degradation, is related to inflammation and allergic responses [10]. It has been reported that, during inflammation and tissue damage, hyaluronidase cleaves polysaccharide HA in low molecular weight HA fragments [11], which induce pro-inflammatory immune responses [12].

We identified esculeosides B (EsB), which contains four times the amount of lycopene content, as a major component in juice or canned tomato [13,14]. Concerning its chemical structure, EsB is a rare and naturally occurring compound which is a solanocapsine-type glycoside. Our previous study demonstrated how EsB ameliorated mice experimental atopic dermatitis by a decrease in serum IgE levels [15]. However, the underlying anti-inflammatory mechanisms are unknown.

Therefore, the present study tested the effects of EsB on hyaluronidase activity and mouse CD4+ T lymphocyte activation. The results demonstrated how EsB and its sapogenol esculeogenin B (Esg-B) inhibited hyaluronidase activity, and suppressed immune activation in mouse CD4+ T cells following in vitro stimulation with concanavalin A (ConA).

## 2. Materials and Methods

### 2.1. Preparation of EsB and Esg-B

EsB was extracted [14]; 900 g of commercially processed tomato juice (*Solanum lycopersicum* L.) was centrifuged. The filtrate passed through a highly porous polystyrene gel (Diaion^®^ HP-20, Mitsubishi Chemical, Tokyo, Japan). The resulting methanol residue was then subjected to dextran gel column (Sephadex^®^ LH-20, Sigma-Aldrich, St. Louis, MO, USA) chromatography and afterwards eluted with 90% methanol. The 90% methanol eluates lacked esculeoside A (EsA) but contained EsB on TLC with CHCl_3_-MeOH-H_2_O (6:4:1). Next, the obtained tomato saponin was identified by the NMR spectrum. The average yield of EsB from commercially produced tomato juice was calculated to be about 0.041%. After a solution of EsB (350 mg) in 2 N HCl was refluxed for 1 h, its reaction mixture was neutralized with 2 N KOH and water was added; next it was passed through Diaion HP-20 and washed with water, then eluted with methanol to give a sapogenol. Crude sapogenol was purified on silica gel column chromatography with CHCl_3_-MeOH-H_2_O (9:1:0.1) to afford the sapogenol Esg-B (25 mg). The chemical structure of EsB is shown in Figure 1A.

### 2.2. Hyaluronidase Inhibitory Assay

The inhibitory activity on hyaluronidase was measured according to a modified Morgan–Elson method as described previously [16,17]. Briefly, samples prepared in 0.1 M acetate buffer (pH 4) and hyaluronidase (type IV-S: From bovine testes, Sigma, St. Louis, MO, USA) in buffer with a final concentration of 5 mg/mL were incubated at 37 °C. Then, compound 48/80 (Sigma) in buffer with a final concentration of 0.5 mg/mL was added and incubated. After hyaluronic acid sodium salt (from rooster comb, Wako, Osaka, Japan) in buffer with a final concentration of 0.4 mg/mL had been included, the mixture was incubated. p-Dimethylaminobenzaldehyde (Wako) acetate solution was then added and incubated. Then the absorbance was determined at 585 nm, and the enzyme inhibitory activity (%) was quantified. EsB or Esg-B as a stock solution (2.8 mg/mL) was prepared with 0.7% DMSO and diluted with 0.1 M acetate buffer. Disodium chromoglycate (Wako) and suplatast tosylate (TCI, Tokyo, Japan) were prepared with water as a stock solution of 10 mg/mL.

### 2.3. Animals and Splenocyte Isolation

The study was submitted to and approved by the Ethics Committee of Sojo University (2019-P-025, 2020-P-025). All experiments were conducted in strict accordance with the Guidelines of the Japanese Pharmacological Society for the Care and Use of Laboratory Animals. Female BALB/c mice, 6~9 weeks old, were obtained from Japan SLC (Hamamatsu, Japan). The animals were housed under conditions of controlled room temperature (24.5–25.0 °C), 60 ± 10% humidity, and a 12/12 h light/dark cycle. Food pellets and water were provided ad libitum. Mice were sacrificed by exposure to isoflurane; the spleen was dissected out and immediately immersed in PBS, then minced and passed through a 70 μm cell strainer. After treatment with RBC lysis buffer to deplete red blood cells, splenocytes were suspended in an RPMI 1640 medium (Wako, Osaka, Japan) supplemented with 10% (*v/v*) fetal bovine serum (FBS), 100 U/mL penicillin, and 100 μg/mL streptomycin (Invitrogen, Waltham, MA, USA) at a density of 3 × 10^6^ cells/mL in 96-well or 24-well plates, and cultured under a humidified atmosphere of 5% CO_2_ at 37 °C before being subjected to various treatments.

### 2.4. Cytotoxicity Assay

Splenocytes were exposed to EsB (2.4, 8, 24, 80, 240  μM) or Esg-B (4, 8, 16, 32, 64 μM) and cultured for 48 h. The live cell rate was examined by an MTT assay and the optical density was read at a wavelength of 570 nm using an Infinite^®^ 200 PRO microplate reader (Tecan Group Ltd., Seestrasse, Switzerland). 

### 2.5. ConA Proliferation Assay

Splenocyte proliferation was labeled using 5-(6)-carboxyfluorescein diacetate succinimidyl ester (CFSE) (Sigma-Aldrich, St. Louis, MO, USA) [18]. Splenocytes (1 × 10^7^ cells/mL) were suspended in 5 μM CFSE (5 mM stock solution in DMSO) in PBS and incubated at 37 °C for 10 min. The labeling process was quenched by adding an equal volume of heat-inactivated FBS. After being washed twice, recounted, adjusted to a density of 5 × 10^5^ cells/mL and seeded onto 24-well plates, CFSE-labeled cells were treated with EsB (10, 30, 100, 300 μM) or Esg-B (1, 3, 10, 30 μM) and stimulated with 1 μg/mL ConA (Sigma-Aldrich), then cultured for 3 days. After harvesting, cells were pre-treated with mouse anti-rat CD16/32 (eBioscience) to block nonantigen-specific binding of immunoglobulins, then incubated at 4 °C for 30 min with phycoerythrin (PE)-conjugated anti-CD3 (eBioscience). CFSE-labeled cells were suspended in the staining buffer, and proliferation was analyzed using BD AccuriTM Plus Flow Cytometer (BD Biosciences, Franklin Lakes, NJ, USA).

Splenocyte proliferation was also analyzed by WST (water-soluble tetrazolium) assay using a Cell Count Kit-8 (CCK-8) (Dojindo, Kumamoto, Japan). After seeding, splenocytes were pretreated with EsB or Esg-B, stimulated with ConA, and cultured for 3 days. Next, the cells were incubated with CCK-8 for 5 h, and the absorbance was recorded using a micro-plate reader at 450 nm. The proliferation rate was compared with that of the untreated control group.

### 2.6. Enzyme-Linked Immunosorbent Assay (ELISA)

Splenocytes were treated with EsB (10, 30, 100 μM) or Esg-B (1, 3, 10 μM) for 1 h before stimulation with ConA (1 μg/mL). After 48 or 72 h, cell-free culture supernatants were collected and stored at − 80 °C until use. The secretion of IL-2, IL-4, IL-10, and IFN-γ were quantified using the corresponding mouse ELISA kits (eBioscience) according to the manufacturer’s instructions.

### 2.7. Quantitative Polymerase Chain Reaction (q-PCR) Assay

Quantification of mRNA expression in splenocytes was performed using RT-PCR by subjecting the cDNA obtained from the following preparation methods to PCR amplification using a StepOnePlus™ Real-Time PCR System (Life Technologies, Waltham, MA, USA).

Total RNA from mouse splenocytes was isolated from homogenates using an RNeasy Mini Kit (Qiagen, Düsseldorf, Germany). The obtained mRNA was quantified by measuring the absorbance at 260 nm, and its quality was determined by measuring the 260/280 ratio. cDNA was synthesized using a PrimeScriptTM 1st strand cDNA Synthesis kit (Takara Bio, Kusatsu, Japan). In brief, 1.0 μg of total RNA from each sample was added to a mixture of 1.0 μL of random 6 mers (50 μM), 1.0 μL of dNTP mix (10 mM), and 8.0 μL of RNase-free dH_2_O, and then held at 65 °C for 5 min and cooled to 4 °C using a thermocycler (Eppendorf, Hamburg, Germany). Next, the denatured mixture was added with 4.0 μL of 5× PrimeScript buffer, 0.5 μL of RNase inhibitor, 1.0 μL of PrimeScript RTase, and 4.5 μL of RNase-free dH2O, and then held at 50 °C for 45 min, heated to 95 °C for 5 min and cooled to 4 °C. Finally, the 20-μL PCR reaction mixture contained 1.2 μL of 10 M forward primers and 1.2 μL of 10 M reverse primers (300 nM final concentration of each primer), 10 μL of PowerUpTM SYBR^®^ Green Master mix (2×), 5.6 μL of nuclease-free water, and 2.0 μL of the obtained cDNA. Thermal cycling conditions were applied as follows: 50 °C for 2 min, 95 °C for 2 min, followed by 40 cycles of 15 s at 95 °C and 1 min at 60 °C. The samples were matched to a standard curve generated by amplifying serially diluted products using the same real-time PCR conditions. The data are presented as the fold change (2^−ΔΔCt^) in gene expression levels and normalized to the mRNA expression levels of an endogenous reference gene, GAPDH (glyceraldehyde 3-phosphate dehydrogenase), then shown relative to that of the control group [19].

The primer sequences of the target genes, selected from PubMed and other databases, were as follows (Table 1):

### 2.8. Statistical Analysis

The data are presented as the mean ± SEM of at least three independent experiments and analyzed by Prism 8 (GraphPad Software, San Diego, CA, USA). Student’s T-test was used to determine statistical significance between two groups, and one-way ANOVA with Tukey’s multiple comparison test was used for multiple groups. *p*-values < 0.05 were considered to be significantly different.

## 3. Results

### 3.1. Hyaluronidase Inhibition by EsB/Esg-B

Firstly, the effects of EsB/Esg-B, chromoglycate, and suplatast on hyaluronidase activity were determined in vitro. As shown in Figure 1B, EsB/Esg-B inhibited hyaluronidase activity in a concentration-dependent manner, and IC50 value was about 200 μM. Chromoglycate induced concentration-dependent inhibition on hyaluronidase with about 10 μM of IC50. Suplatast did not inhibit hyaluronidase activity.

### 3.2. Cytotoxicity of EsB/Esg-B in Splenocytes

Our previous study demonstrated that the oral administration of EsB at 10 mg/kg or fresh tomato fruits saponin EsA at 10–100 mg/kg ameliorated experimental dermatitis in mice [15]. It has been reported that esculeogenin A (Esg-A), a sapogenol of EsA, at 3-100 μM inhibited the accumulation of cholesterol ester in human monocyte-derived macrophages [20]. Considering these in vivo and in vitro effects, various concentrations of EsB and Esg-B were tested for cytotoxicity to splenocytes as shown in Figure 2. When the live cell rate of the control was normalized to 1.0, the ratios of live splenocytes were 1.01, 0.96, 1.04, 0.92, and 0.70 in the presence of EsB at 2.4, 8, 24, 80, and 240 μM, and 1.05, 1.10, 1.04, 0.84, and 0.62 in the presence of Esg-B at 4, 8, 16, 32, and 64 μM, respectively. EsB at about 150 μM induced the cytotoxicity by 20%, and that by Esg-B is about 35 μM.

### 3.3. Suppression of T Lymphocyte Proliferative Response by EsB/Esg-B

Effects of EsB/Esg-B on T lymphocyte proliferative response were tested using in vitro ConA-stimulated T cell proliferation (Figure 3). T lymphocyte proliferative effect in the culture with EsB/Esg-B was compared to that without EsB/Esg-B (ConA alone). As shown in Figure 3A,B, using the WST assay, addition of EsB/Esg-B to the culture system resulted in a concentration-dependent suppression of the proliferation response. Esg-B at about 5 μM inhibited the elevated proliferation ratio by 50%, whereas EsB at about 120 μM was required to show the same inhibition.

In Figure 4, further assays using CD3-PE+/CFSE+ fluorescence demonstrated whether or not this dose-dependent suppression by EsB/Esg-B was related to T cell division. Daughter T cells (Figure 4A, R2), derived from responder splenocytes following ConA stimulation, were differentiated from undivided T cells (R1) and identified by the intensity of CFSE staining (Figure 4A,B) using flow cytometry. The addition of EsB/Esg-B to CFSE-labeled cells stimulated with ConA showed a tendency to suppress T lymphocyte division (Figure 4B,C). Esg-B at about 3 μM decreased dividing T cells by 50%, whereas about 100 μM EsA was needed to show the same reduction. 

### 3.4. EsB/Esg-B Modulation of ConA Blast-Induced Cytokine Level

During T cell activation, cytokines are produced to modulate the differentiation and subsequent specialization of T cells; thus, we examined the effects of EsB/Esg-B on ConA blast-induced cytokine extracellular secretion. EsB/Esg-B decreased the production of Th2 cytokine IL-4 and the Th1 cytokine IFN-γ in a concentration-dependent manner, as shown in Figure 5A. In detail, Esg-B at about 2 μM inhibited the elevated IL-4 and IFN-γ production by 50%, whereas EsB at about 30 μM and 20 μM was required to show the same inhibition, respectively, indicating the similar inhibitory effect of EsB/Esg-B on Th2 and Th1 cytokine production. Further, the inhibitory effect of EsB/Esg-B on Th2/Th1 cytokine production is greater than for IL-2 production. Moreover, EsB/Esg-B also decreased production of the Treg cytokine IL-10, which are required for Treg cell maintenance. In detail, Esg-B at about 1 μM inhibited the elevated IL-10 production by 50%, whereas EsB at about 10 μM was required to show the same inhibition.

We also examined the effect of EsB/Esg-B on ConA blast-stimulated cytokine mRNA expression. EsB/Esg-B decreased mRNA expression levels of IL-4 and IFN-γ in a concentration-dependent manner, as shown in Figure 5B. In detail, Esg-B at both 10 μM inhibited the elevated IL-4 and IFN-γ gene expression levels by 50%, whereas EsB at both 100 μM were required to show the same inhibition, respectively, also indicating the similar inhibitory effect of EsB/Esg-B on Th2 and Th1 gene expression level.

## 4. Discussion

In this in vitro study, it showed that the saponin EsB and its sapogenol Esg-B from tomato juice alleviate ConA-blast T lymphocyte activity by modulation of Th1/Th2/Treg-associated cytokines signaling.

Our previous study showed that oral administration of EsB, a solanocapsine-type glycoside and a major component in tomato juice, ameliorated experimental dermatitis in mice through decreases in IgE and ConA-mitogenic action, and a decline in IL-4 secretion [14,15]. However, the underlying immunologic molecular mechanisms are unknown. Hence, the present study investigated the effects of EsB/Esg-B on ConA-blast mouse splenocytes, and analyzed the immune mechanism in relation to CD4+ T lymphocytes. We pretested the EsB/Esg-B cytotoxic effects in mouse primary splenocytes, and found that cytotocity was lower at the concentrations of EsB < 150 μM and Esg-B < 30 μM (Figure 2). Based on these concentrations, we first tested the effects of EsB/Esg-B on T lymphoproliferative action induced by ConA, a selective T cell mitogen. Using WST assay, EsB/Esg-B suppressed splenocyte proliferation (Figure 3), and showed profound suppression of CD3+ T lymphocyte division by flow cytometry (Figure 4). Both assays showed that EsB/Esg-B decreased T lymphocyte proliferation; however, the latter may suggest their potential effect on T lymphocytes. The present T lymphoproliferative decline by EsB is in line with our previous in vivo study using EsB [15].

We subsequently tested EsB/Esg-B for their Th2/Th1 cytokine production-modulatory potential, since in the atopy patch test it was observed that T cells in the skin display an initial Th2 polarization, with increasing populations of Th1 cells in patients with chronic AD [21,22,23]. The high proportion of Th2-polarized T cells appears to be a key factor in patients with allergic inflammation [24]. The present study used the ConA-stimulated lymphoproliferative system, and it was found that EsB/Esg-B suppressed both Th2/Th1 cytokine production and intracellular gene expression levels (Figure 5). Moreover, EsB/Esg-B inhibitory degrees on Th2-mediated cytokine production and gene expression were similar to those on Th1-mediated responses. Recently, the clinical efficacy of the IL-4 receptor antagonist dupilumab was demonstrated, in addition to a decline in Th1 and Th2 cell numbers [25,26], supporting evidence for the importance of the Th2/Th1 immune pathway in AD. Thus, we highlight that EsB/Esg-B can suppress Th2/Th1 dominant inflammatory reaction.

In addition to Th2 and Th1 effector cells, CD4+ Th cells are controlled by Treg cells, which regulate inflammatory responses and restore immune homeostasis. We examined the Treg-modulatory potential of EsB/Esg-B, and demonstrated that EsB/Esg-B suppressed Treg cytokine IL-10 production (Figure 5). This indicated that EsB/Esg-A alleviated the suppressive action of IL-10-producing Treg cells in mouse splenocytes in response to ConA mitogenic stimuli. At the same time, Esg-B modulation in CD4+ Th cells is characterized by a reduction in IL-2 signaling, including IL-2 cytokine secretion (Figure 5). Since IL-2 has been reported as an important activator of Treg suppressive activity in vitro and in vivo [27,28,29,30], and the present data suggests that during suppressed IL-2 signaling by Esg-B, the concomitant suppression of Treg and/or the other immune cells may still counteract ConA-induced in vitro activation of cellular immunity. On the other hand, EsB did not affect IL-2 production, a point that needs to be examined further.

Moreover, this study also indicates that the inhibitory effect of EsB/Esg-B on mammalian hyaluronidase activity is weaker than that of chromoglycate, an anti-allergic agent, known as a mast cell stabilizer, which inhibits the release of mediators, such as histamine and leukotrienes, and is stronger than that of suplatast, another anti-allergic agent, which is able to alleviate cutaneous symptoms of atopic dermatitis and nasal symptoms of allergic rhinitis by inhibiting the production of Th2 cytokine and immunoglobulin E [31,32], and which also suppresses histamine release and H1 receptor expression [33]. Thus, further research in chemical mediator release is warranted for molecular targets of anti-allergic therapeutic effects of EsB/Esg-B. Moreover, their inhibitory activity on the hyaluronidase activity might be not only in managing atopic dermatitis, but also in approaching intestinal diseases, exactly through hyaluronic acid degradation in the intestinal bio-film.

Finally, in a comparison of EsB- and Esg-B-suppressive potential in Th2/Th1/Treg activation, the suppression by Esg-B is greater than that by EsB, suggesting that the steroidal alkaloid Esg-B moiety may be mainly responsible for the suppressive effect on CD4+ T cell activity at least in vitro. The main effect of Esg-B should be consistent with our previous report, in which the final metabolites were eliminated as androsterone analogues in the urine of men orally administered ripe cherry tomato fruit [34].

## 5. Conclusions

Taken together, these data indicate that tomato juice saponin EsB and its sapogenol Esg-B are capable of down-regulating in vitro ConA-mitogenic CD4+ T lymphocyte activation via modulating the differentiation of Th2, Th1, and Treg subsets, thereby contributing to the EsB-mediated alleviation of experimental dermatitis in mice. Next steps include investigation of their downstream stimuli and the signaling pathways affecting the Th2/Th1/Treg cells. This section is not mandatory but can be added to the manuscript if the discussion is unusually long or complex.

## Figures and Tables

**Figure 1 jpm-12-00579-f001:**
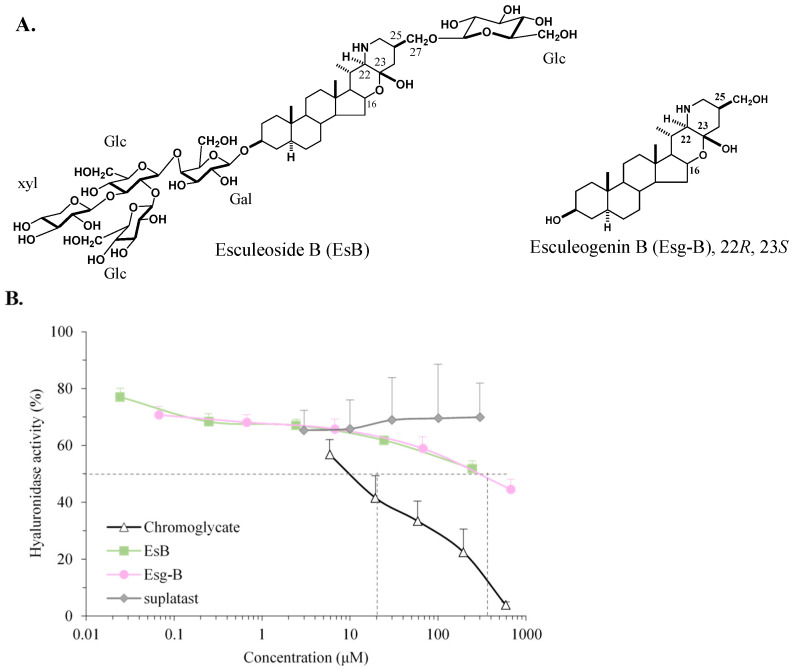
Chemical structures and hyaluronidase inhibition by EsB/Esg-B. (**A**) The chemical structures of EsB and Esg-B. Glc: Glucose; Gal: Galactose; Xyl: Xylose. Molecular weight: 1228.34 g/mol (EsB); 447.33 g/mol (Esg-B). (**B**) Dose-dependence of EsB/Esg-B on the inhibition of hyaluronidase activity. The results represent five independent experiments, and each bar is expressed as mean ± SEM.

**Figure 2 jpm-12-00579-f002:**
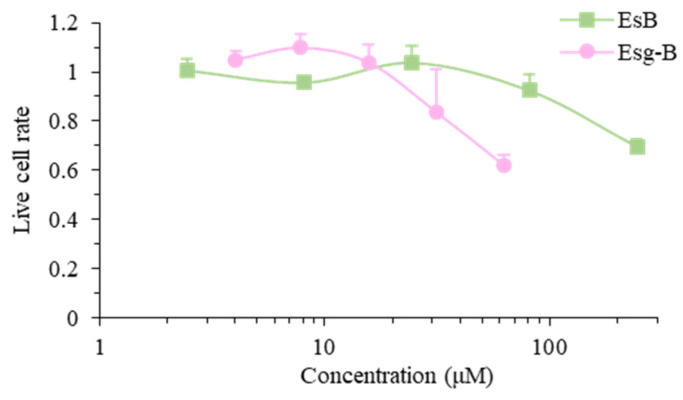
Cytotoxicity of EsB/Esg-B in splenocytes. Splenocytes (3 × 10^6^ cells/well) were seeded onto 24-well plates and treated with 0.1% DMSO, or the indicated concentrations of EsB or Esg-B for 48 h. The live cell rate was examined by an MTT assay. The results represent four independent experiments, and each bar is expressed as mean ± SEM.

**Figure 3 jpm-12-00579-f003:**
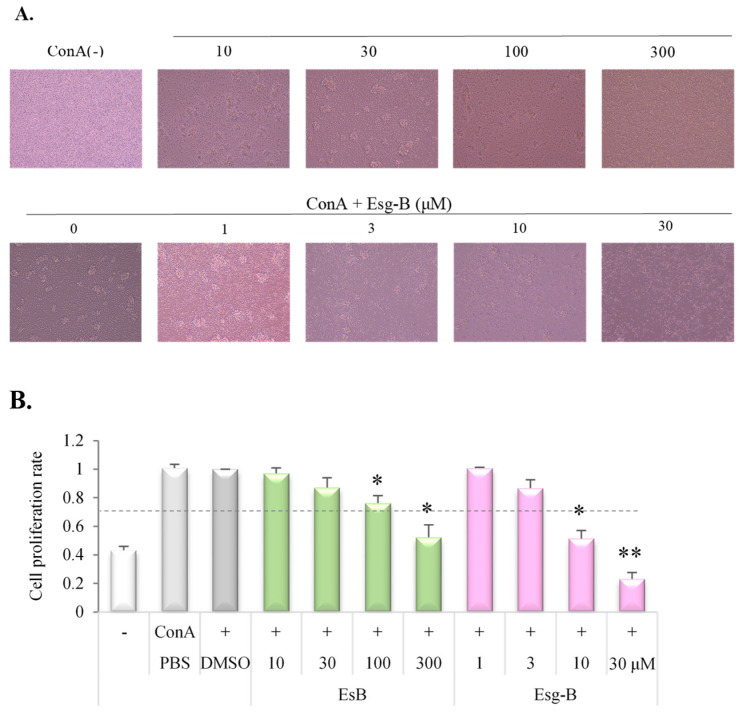
Inhibition of ConA blast proliferation by EsB/Esg-B using WST assay. (**A**) Photographs show ConA blast lymphocyte and the inhibition by EsB/Esg-B. (**B**) Expression of cell proliferation ratio of ConA blasts without and with EsB/Esg-B addition was determined by the WST assay. Splenocytes were treated with 0.1% DMSO or the respective concentrations of EsB/ Esg-B for 1 h, then stimulated with ConA (1 μg/mL) for 48 h. Results were normalized to the optical density of the culture by ConA alone. Data represent five independent experiments, and each bar is expressed as mean ± SEM. *: *p* < 0.05, **: *p* < 0.01, significantly different from the control (ConA alone). EsB: Esculeoside B; Esg-B: Esculeogenin B. The grey dotted line shows about 50% of ConA-induced elevated responses.

**Figure 4 jpm-12-00579-f004:**
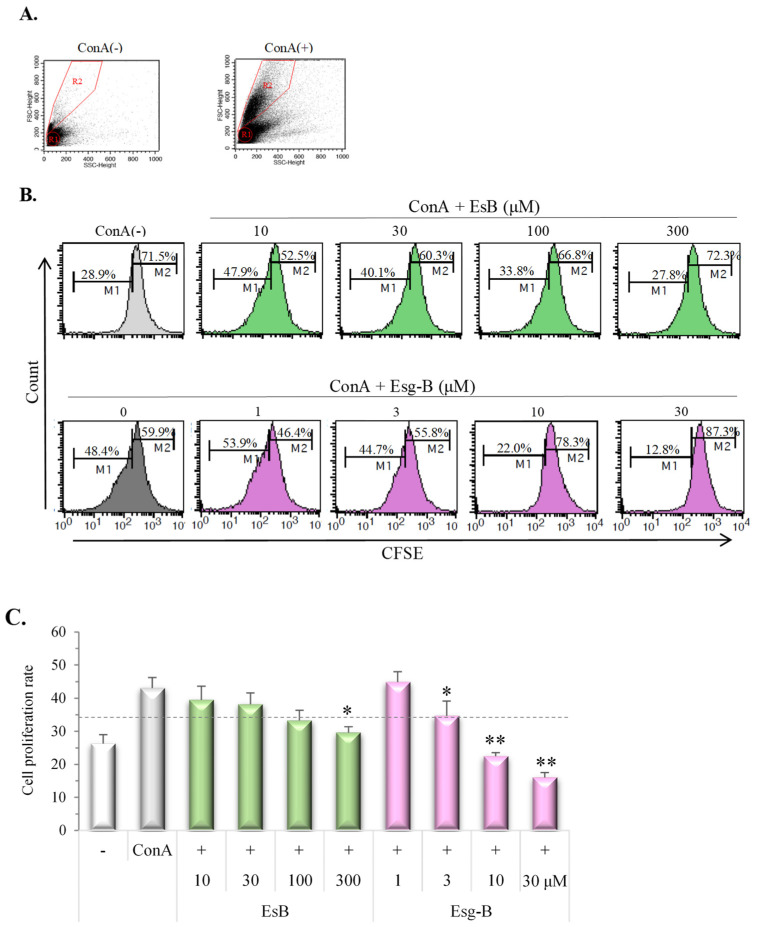
EsB/Esg-B inhibition of ConA blast proliferation tracing by CFSE. (**A**) Cell division in CFSE-labeled splenocytes was determined by flow cytometric analysis. Splenocytes were labeled with CFSE and treated with 0.1% DMSO, or the respective concentrations of EsB/Esg-B for 1 h, then stimulated with ConA (1 μg/mL) for 72 h. Alterations in light scatter characteristics: splenocytes alone (left), ConA blast of splenocytes (right). All plots were gated on CD3-positive cells including both resting lymphocytes (R1) and blasts (R2). (**B**) Representative histogram plots show the cell division associated with CFSE-labeled splenocytes without and with EsB/Esg-B, while the histograms were gated to both R1 and R2. (**C**) Expression of cell division percentage (M1) by ConA blast without and with EsB/Esg-B addition. Data represent five independent experiments, and each bar is expressed as mean ± SEM. *: *p* < 0.05, **: *p* < 0.01, significantly different from the Control (ConA alone). EsB: Esculeoside B; Esg-B: Esculeogenin B. The grey dotted line shows about 50% of ConA-induced elevated responses.

**Figure 5 jpm-12-00579-f005:**
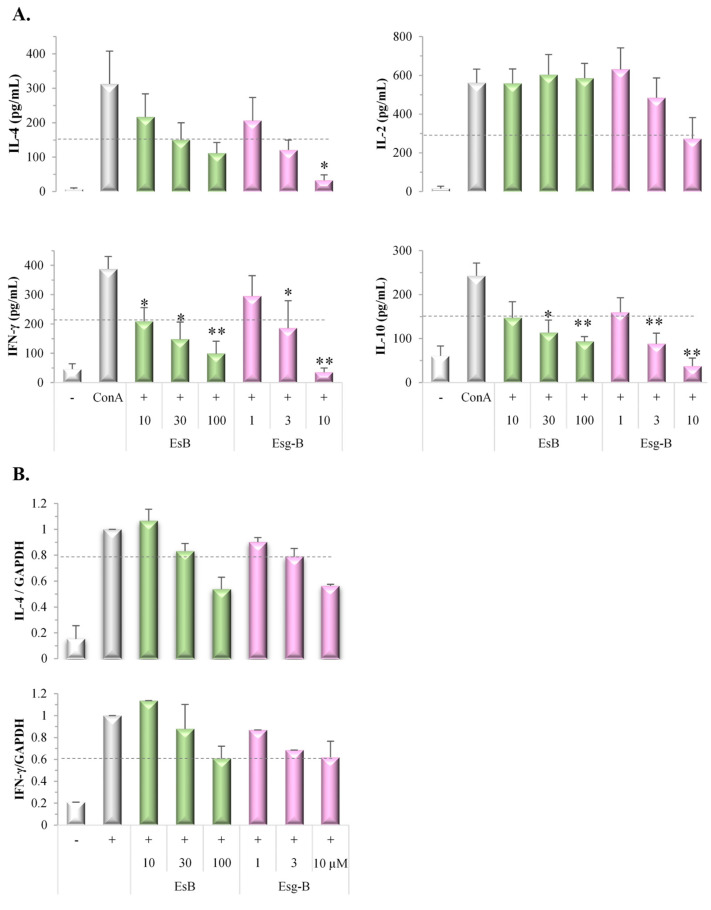
Modulation of ConA blast-induced cytokine production and gene expression by EsB/Esg-B. Splenocytes (3 × 10^6^ cells/well) were pretreated with 0.1% DMSO or the respective concentrations of EsB/Esg-B for 1 h, and then stimulated with ConA (1 μg/mL) for 48 h or 72 h. The culture supernatants were collected and assayed for cytokine secretion using ELISA. The cells were harvested, and RNA was extracted and reverse transcribed to cDNA. Cytokine mRNA expression was determined by RT-PCR. (**A**) EsB/Esg-B decreased IL-4, IL-2, IFN-γ, and IL-10 production in ConA-stimulated splenocytes. Data represent three independent experiments. (**B**) EsB/Esg-B decreased IL-4 and IFN-γ mRNA expression in ConA-stimulated splenocytes. The relative mRNA expression was normalized to the endogenous control gene GAPDH and calibrated using EsB/Esg-B untreated (ConA alone) cells. Data represent three independent experiments. Each bar is expressed as mean ± SEM. *: *p* < 0.05, **: *p* < 0.01, significantly different from the control (ConA alone). EsB: Esculeoside B; Esg-B: Esculeogenin B. The grey dotted line shows about 50% of ConA-induced elevated responses. Authors should discuss the results and how they can be interpreted from the perspective of previous studies and of the working hypotheses. The findings and their implications should be discussed in the broadest context possible. Future research directions may also be highlighted.

**Table 1 jpm-12-00579-t001:** Quantitative polymerase chain reaction primers for GAPDH, IL-4, IFN-γ, mRNA.

	Forward	Reverse
GAPDH	5′-CCCAGCAAGGACACTGAGCAAG-3′	5′-GGTCTGGGATGGAAATTGTGAGGG-3′
IL-4	5′-GAAGCCCTACAGACGAGCTCA-3′	5′-ACAGGAGAAGGGACGCCAT-3′
IFN-γ	5′TCTGGGCTTCTCCTCCTGCGG-3′	5′GGCGCTGGACCTGTGGGTTG-3′

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
