# Peer review of "Decrease of Hyaluronidase Activity and Suppression of Mouse CD4+ T Lymphocyte Activation by Tomato Juice Saponin Esculeoside B, and Its Sapogenol Esculeogenin B"

_jpm, 2022, doi:10.3390/jpm12040579_

Round 1

Reviewer 1 Report

Sapogenols from tomato varieties, and other vegetal sources are mainly known for their anti-hyperlipidemic and anti-atherosclerosis effects. Also demonstrating their inhibitory activity on the hyaluronidase activity might be of high medical usefulness not only in managing atopic dermatitis, but also in approaching intestinal diseases, exactly through hyaluronic acid degradation in the intestinal bio-film.

Author Response

Dear Reviewer,

MDPI JPM Editorial Office

We address the following points, and have submitted a redline version of our manuscript to MDPI JPM.

We sincerely thank your pertinent advices in improving our manuscript.

Kind regards,

Jian-Rong Zhou

Department of Presymptomatic Medical Pharmacology

Faculty of Pharmaceutical Sciences, Sojo University, Japan

Comments and Suggestions for Authors

Sapogenols from tomato varieties, and other vegetal sources are mainly known for their anti-hyperlipidemic and anti-atherosclerosis effects. Also demonstrating their inhibitory activity on the hyaluronidase activity might be of high medical usefulness not only in managing atopic dermatitis, but also in approaching intestinal diseases, exactly through hyaluronic acid degradation in the intestinal bio-film.

We added your suggestion in Line 419-422.

Reviewer 2 Report

Recommendation: Reconsider after major revisions

Comments:

The submitted manuscript entitled “Decrease of hyaluronidase activity and suppression of mouse CD4+ T lymphocyte activation by tomato juice saponin esculeoside B, and its sapogenol esculeogenin B” describes the compounds esculeoside B and esculeogenin B with their immunologic molecular mechanisms. The methods are clearly described and the paper is of interest to the readers of Journal of Personalized Medicine and I would recommend reconsider the manuscript after the comments are addressed.

Major Corrections:

There is no figure present in the manuscript, only the legends. Please add the figures in.

Some references have the doi information while the rest have not. Please keep them consistent.

Line 382: Please double check this reference, delete it or rearrange it.

Other comments:

  • Line 29: WST-assay should be the full name since it is the first time shown in the manuscript.
  • Line 33: “Our study is the first to” could be revised as “Our study is the first time to”
  • Line 81: Please report how much EsB was isolated from the 900 g juice. What is the meaning of “the average of Esb…”? Please correct it.
  • Line 87: …afford the sapogenol Esg-B (25 mg).
  • Line 101: 10 mg/mL
  • Line 200: Please double check “at 2.4, 8, 24, 80 and 240 μM”
  • Line 201: the decimal digit of 1.1 should be consistent with the others.
  • Line 241: “it is shown that” should be corrected as “it showed that”
  • Line 309: Esg-A should be corrected as Esg-B.
  • Line 310: “HA, hyaluronic acid”

Author Response

Dear Reviewer,

MDPI JPM Editorial Office

We address the following points, and have submitted a redline version of our manuscript to MDPI JPM.

We sincerely thank your pertinent advices in improving our manuscript.

Kind regards,

Jian-Rong Zhou

Department of Presymptomatic Medical Pharmacology

Faculty of Pharmaceutical Sciences, Sojo University, Japan

Comments and Suggestions for Authors

The submitted manuscript entitled “Decrease of hyaluronidase activity and suppression of mouse CD4+ T lymphocyte activation by tomato juice saponin esculeoside B, and its sapogenol esculeogenin B” describes the compounds esculeoside B and esculeogenin B with their immunologic molecular mechanisms. The methods are clearly described and the paper is of interest to the readers of Journal of Personalized Medicine and I would recommend reconsider the manuscript after the comments are addressed.

Major Corrections:

There is no figure present in the manuscript, only the legends. Please add the figures in. Some references have the doi information while the rest have not. Please keep them consistent.

 We have added the figures in the manuscript.

We have added the doi information in Line 470, 472, 475, 476, 482, 484, 485, 490, 492, 494, 496, 499, 508, 510, 517, 519, 530, 532, 538, 543 and 545.

Line 382: Please double check this reference, delete it or rearrange it.

We deleted the latter of this reference.

Other comments:

  • Line 29: WST-assay should be the full name since it is the first time shown in the manuscript.

We added (water-soluble tetrazolium) in Line 19.

  • Line 33: “Our study is the first to” could be revised as “Our study is the first time to”

time is added in Line 22.

  • Line 81: Please report how much EsB was isolated from the 900 g juice. What is the meaning of “the average of Esb…”? Please correct it.

We corrected as “The average yield of EsB……” in Line 68. The average yield of EsB is about 0.041 g from 1000 g tomato juice.

  • Line 87: …afford the sapogenol Esg-B (25 mg).

It is corrected in Line 74.

  • Line 101: 10 mg/mL

It is corrected in Line 89.

  • Line 200: Please double check “at 2.4, 8, 24, 80 and 240 μM”

It is OK in Line 191.

  • Line 201: the decimal digit of 1.1 should be consistent with the others.

   It is corrected as 1.10 in Line 191.

  • Line 241: “it is shown that” should be corrected as “it showed that”

    It is corrected in Line 313.

  • Line 309: Esg-A should be corrected as Esg-B.

It is corrected in Line 439.

  • Line 310: “HA, hyaluronic acid”

It is corrected in Line 440.

Round 2

Reviewer 2 Report

Recommendation: Accept after minor revisions

Comments:

The resubmitted manuscript entitled “Decrease of hyaluronidase activity and suppression of mouse CD4+ T lymphocyte activation by tomato juice saponin esculeoside B, and its sapogenol esculeogenin B” describes the compounds esculeoside B and esculeogenin B with their immunologic molecular mechanisms. The authors have addressed the comments. I would recommend the manuscript be accepted after the minor revisions.

Comments:

  • Line 19: Correct it as “water-soluble tetrazolium (WST)-assay”.
  • Line 69: The average yield of EsB is about 0.041 g from 1000 g tomato juice from the authors’ reply. Please confirm the calculated yield of 0.041%. Maybe 0.0041%?
  • Line 176: In Figure 1A, there should be two structures to present EsB and Esg-B, respectively. In addition, the drawing of the stereochemistry of the aglycon should be refined.
  • Line 299: Some extra spaces should be removed.

Author Response

Dear Reviewer,

MDPI JPM Editorial Office

We address the following points, and have submitted a redline version of our manuscript to MDPI JPM.

We sincerely thank your pertinent advices in improving our manuscript.

Kind regards,

Jian-Rong Zhou

Department of Presymptomatic Medical Pharmacology

Faculty of Pharmaceutical Sciences, Sojo University, Japan

...................................................................................................

Comments:

The resubmitted manuscript entitled “Decrease of hyaluronidase activity and suppression of mouse CD4+ T lymphocyte activation by tomato juice saponin esculeoside B, and its sapogenol esculeogenin B” describes the compounds esculeoside B and esculeogenin B with their immunologic molecular mechanisms. The authors have addressed the comments. I would recommend the manuscript be accepted after the minor revisions.

 Comments:

Line 19: Correct it as “water-soluble tetrazolium (WST)-assay”.

It is corrected.

Line 69: The average yield of EsB is about 0.041 g from 1000 g tomato juice from the authors’ reply. Please confirm the calculated yield of 0.041%. Maybe 0.0041%?

I am sorry, I made a mistake. The average yield of EsB is about 0.041 g from 100 g tomato juice not from 1000 g.

Line 176: In Figure 1A, there should be two structures to present EsB and Esg-B, respectively. In addition, the drawing of the stereochemistry of the aglycon should be refined.

It is corrected.
